# Cyclic Deformation Behavior of Additive-Manufactured IN738LC Superalloys from Virgin and Reused Powders

**DOI:** 10.3390/ma15248925

**Published:** 2022-12-14

**Authors:** Jialiang Chen, Jinghao Xu, Mikael Segersäll, Eduard Hryha, Ru Lin Peng, Johan Moverare

**Affiliations:** 1Division of Engineering Materials, Department of Management and Engineering, Linköping University, SE-58183 Linköping, Sweden; 2Division of Materials and Manufacture, Department of Industrial and Materials Science, Chalmers University of Technology, SE-41296 Gothenburg, Sweden

**Keywords:** additive manufacturing (AM), powder reuse, laser powder bed fusion (L-PBF), IN738LC superalloy, fatigue, fractography

## Abstract

In laser powder bed fusion (L-PBF), most powders are not melted in the chamber and collected after the printing process. Powder reuse is appreciable without sacrificing the mechanical properties of target components. To understand the influences of powder reuse on mechanical performance, a nickel-based superalloy, IN738LC, was investigated. Powder morphology, microstructure and chemical compositions of virgin and reused powders were characterized. An increase in oxygen content, generally metallic oxides, was located on the surface of powders. Monotonic tensile and cyclic fatigue were tested. Negligible deterioration in strength and tensile ductility were found, while scattered fatigue performance with regard to fatigue life was shown. Deformation and fatigue crack propagation mechanisms were discussed for describing the powder degradation effects.

## 1. Introduction

Nickel-based superalloys are widely used as critical components in high-temperature regions of gas turbines, such as turbine blades and vanes, owing to their excellent combination of high-temperature mechanical and chemical properties [1]. The modern nickel-based superalloys have been well moderated into dual phase form, with the ordered L1_2_ γ′ intermetallic phase acting as the primary precipitation strengthening source, which is embedded in the continuous disordered FCC γ nickel matrix. In recent decades, the rapid development of additive manufacturing (AM), also known as three-dimensional (3D) printing, has offered advantages of flexibility to design engineers with the layer-by-layer manufacturing process [2]. Powder bed fusion (PBF) is one of the most common AM techniques to fabricate metallic materials based on the localized fusion of the pre-alloyed metallic powder bed via the direct energy inputs [3]. Among the PBF processes, laser-PBF (L-PBF) and electron-beam-PBF (EB-PBF) are the two major types of methods to produce the engineering metallic materials [4]. Generally, the L-PBF is conducted under an inert gas atmosphere, such as argon. After each printing process, the unmolten and unfused loose powder is collected, sieved and reused for the following prints to maximize powder feedstock utilization [5]. With the increasing number of powder reuse, there is a risk of the change in powder morphology, especially chemical composition, leaving negative impacts on L-PBF processability and the quality of the produced parts.

The reused pre-alloyed powder decays in various ways. Powder degradation is a general phenomenon during the additive manufacturing process for most alloys. For example, the increase in oxygen content is commonly found in reused powders [6,7,8]. As reported by Gruber et al. [9], the reused powder becomes degraded after varying numbers of reuse cycles, mainly owing to the increment of Al-rich oxides. Oxide particles with high-melting temperature affected the printing integrity. They increased the susceptibility for the formation of lack of fusion defects, which is fatal in both strength and ductility, during printing of the IN718 superalloy from reused powder [10]. 

In addition to the influence of powder reuse on powder characteristics and printing quality, usage of reused powder may also affect the mechanical properties of printed components. As reviewed and summarized by Santecchia et al. [11], no consistent trend could be deduced among a wide batch of metallic materials fabricated by the L-PBF process. Specifically, Ti-6Al-4V alloys [12] exhibited a slight increase in the ultimate tensile strength of the samples fabricated by the reused powder. In other alloys, such as AlSi10Mg, mechanical behaviors, including tensile strength, yield strength, elongation and microhardness, all decreased up to 15% when powders were reused for two and a half years [13]. The reduction of ductility was also observed in IN718 superalloys owing to powder reuse and corresponding higher levels of porosity in the print [14]. A case-by-case study of powder reuse effect is necessary for different alloys, including a comprehensive record of powder reuse procedures.

In addition to tensile properties, cyclic mechanical behavior is vital for most industrial applications. Paccou et al. [15] investigated the effect of powder reuse on the fatigue behavior of surface-machined IN718 by L-PBF. At a relatively large total strain, no clear difference in terms of fatigue life between the samples from new and reused powders was observed, while at the relatively low total strain regime, a slight reduction in fatigue life was observed in the samples from reused powders. In contrast, contradictory fatigue performances of other L-PBF alloys fabricated with reused powder were also reported. In particular, Carrion et al. [16] discovered a longer high cycle fatigue life in surface-machined Ti-6Al-4V alloys printed by reuse powders with improved flowability and lower compressibility. A similar trend in advanced fatigue was also reported by Arash et al. [17] in 17-4PH steel printed by used powders. Nevertheless, both instances determined that the machined surface is necessary because it could eliminate potential surface defects and provide excellent fatigue properties. However, the influence of powder reuse on the fatigue properties of L-PBF alloys has not yet been well investigated.

As a typical γ′ precipitation-strengthened nickel-based superalloy, the study on IN738LC (LC for low carbon) during the L-PBF process is of significant interest. IN738LC materials fabricated by L-PBF have been previously investigated for tensile properties [18,19,20], high-temperature short-term creep performance [21] and hot corrosion behavior [22], focusing on the relationship between microstructure and mechanical properties. However, there has been limited research on the influence of powder reuse on fatigue behaviors. 

This raises the question: what is the influence of powder reuse on the cyclic deformation behavior of the L-PBF-processed IN738LC superalloy? With the aim of understanding the powder reuse effects, two sets of specimens of IN738LC nickel-based superalloy were respectively fabricated from virgin and reused powders in this study. The virgin and reused powders were comprehensively characterized. Identical hot isostatic pressing and surface machining were applied on both specimens before tests to reduce the microstructural variables induced by the printing. Stress-controlled fatigue behaviors were compared. Through the characterization of post-mortem specimens, the deformation and fatigue crack propagation mechanisms were investigated and discussed. 

## 2. Materials and Experimental Procedures

### 2.1. Materials

Specimens of the nickel-based superalloy IN738LC were horizontally built by the L-PBF process on an EOS M290 (EOS GmbH, Krailling, Germany) system equipped with a 400 W Yb-fiber laser. To fabricate the near-dense part, optimized printing conditions were used [23], following the printing parameters listed in Table 1. After printing, HIP at 1210 °C for 4 h was carried out.

The IN738LC gas atomized powder used for the L-PBF process was supplied by EOS Oy, (Turku, Finland). Two powders were studied: a virgin powder and a reused powder which was reused in more than 10 cycles of L-PBF printing. After each build cycle, approximately 20% of the fresh powder was added to ensure the necessary amount of powder for the next building mission. After each L-PBF process, the evacuated powder was sieved using 45 µm mesh to remove coarse particles. The specimens fabricated by virgin powder were denoted as E1, while the specimens fabricated by reused powder were denoted as E2. 

The overall chemical compositions of the as-gas-atomized powder was provided by the supplier, and are listed in Table 2. Measured by means of inert gas fusion (IGF), the oxygen content in virgin powder was 140 ppm compared to 170 ppm in the reused powder.

### 2.2. Mechanical Tests

Room temperature monotonic tensile tests were carried out in an Instron 5582 universal testing system (Instron®, Norwood, MA, USA) for both E1 and E2 specimens. One specimen from each condition was used for the tensile testing. The nominal strain rate was 10^−3^/s under constant displacement control. An extensometer was used to measure the strains up to 2%, which covered the elastic and elasto-plastic regimes. Young’s moduli were calculated by using the stress and extensometer-measured strain values in the elastic region. The yield strength was determined by the 0.2% offset line.

Stress-controlled fatigue tests were conducted in an Instron 8800 servohydraulic testing system (Instron®, Norwood, MA, USA) with a loading capacity of ±50 kN. Fully reversed tension-compression fatigue tests (load-ratio R = −1) at room temperature were performed with a target stress amplitude from 520 to 700 MPa. There were nine specimens produced from virgin powder and eight specimens produced from reused powder for the fatigue tests. The load frequency was 10 Hz. A maximum number of cycles was set at two million cycles for the run-out limit. An extensometer with a gauge length of 12.5 mm was attached to the surface of the fatigue specimen to monitor the total strain. 

Dog-bone-shaped specimens were used for both tensile and fatigue tests, as illustrated in Figure 1a, according to the ASTM E466-15 standard [24]. Before testing, the fatigue bars were polished in parallel to the loading direction to remove influence from the transverse scratches (Figure 1b). After polishing, the measured surface roughness was 0.148 (±0.05) μm Ra (required Ra < 0.2 μm according to ASTM E466-15 standard [24]) along the longitudinal direction. As shown in Figure 1c, the optical micrograph illustrates the pristine microstructure before testing. In this study, all specimens were horizontally built, indicating that the loading direction of both monotonic tensile and cyclic fatigue testing was perpendicular to the build direction (BD) of the L-PBF process.

### 2.3. Microstructure Characterization

Metallographic specimens were polished on a Struers Tegramin-30 sample preparation system with the finish step using colloidal silica suspension. In order to reveal the precipitate, chemical etching was conducted at an ambient temperature using a Marble’s reagent. Fracture surface observations were performed on a Leica M205C stereo optical microscope (OM) (Leica Microsystems GmbH, Wetzlar, Germany) and a Hitachi SU70 (Hitachi, Ltd., Tokyo, Japan) field emission scanning electron microscope (SEM). Electron backscatter diffraction (EBSD) (Oxford Instruments, Oxfordshire, UK) measurements were conducted on the pristine microstructures and cross-sections of the fatigue tested specimens at the crack initiation region. The EBSD data were acquired with an Oxford EBSD detector at an electron beam accelerating voltage of 20 kV. Post-processing of EBSD data was carried out using the open-source MATLAB 2017b package MTEX 5.7.0 [25].

X-ray photoelectron spectroscopy (XPS) was used to identify the elements and their chemical state on the surfaces of both powders. It was carried out on a PHI 5500 (ULVAC-PHI) (ULVAC-PHI, Inc., Hagisono, Japan) instrument equipped with a monochromatic Al Kα (1486.6 eV) X-ray source for the generation of photoelectrons. The measured area was around 0.2 mm in diameter, which indicated that results represented the average chemical state of the analyzed powder surfaces. Surface compound thickness and chemical composition depth were obtained by altering XPS analysis and ion etching (Ar+). The ion etching rate was 5.2 nm/min as calibrated on a flat oxidized tantalum foil. High resolution narrow scans over the binding energy of the elements of interest were collected using a pass energy of 26 eV.

## 3. Results

### 3.1. Powder Characterization

SEM micrographs of the virgin and reused powders are shown in Figure 2a,b, respectively. The morphology of the two different powders appear to be similar. The size and shape of powder particles were measured from three groups of powder containing approximately 400 particles in each group. The statistical results are shown in Figure 2c,d. The circularity is used to characterize the shape of the particles as expressed:(1)$$\begin{array}{c} Circularity=4\pi \cdot \frac{area}{perimeter^{2}}\; \end{array}$$

It is important to note that evaluation of circularity in the case of agglomerates based on 2D imaging is deemed to increase scatter. A circularity value of 1.0 indicates a perfect circle, while near 0 indicates an increasingly elongated shape. The dominant two-dimensional shapes of both powders are close to a circle, as summarized in Figure 2c. The average circularity of virgin and reused powders is 0.80 and 0.82, respectively. According to the circularity results, it is reasonable to assume the powder as spherical particles; then, the size is determined by the powder equivalent diameter. This data is a good fit in terms of Gaussian distribution for the two powders, as illustrated in Figure 2d. The mean diameter (MD) for E1 particles is for E2 particles. There is a slight decrease in mean diameter for reused powder particles compared to virgin powder particles.

Figure 3 shows a detailed view of the representative morphologies of the powders under a higher magnification of SEM micrograph. Again, the majority of powder particles display a near-spherical shape (Figure 3a1,b1). The circularity of certain particles is normally over the average circularity, as indicated in Figure 2c.

However, one can also visually identify non-spherical irregular particles, which are typically agglomerates of the finer powder particles. Generally, such particles consist of one or a couple of coarse particles and fine satellites connected to its surface, which can be found in both virgin (Figure 3a2) and reused (Figure 3b2) powders, and hence, has an origin in powder atomization [26]. They are formed either during atomization [26] or during powder reuse due to the formation of the agglomerates in the powder, thanks to the melted spatter particles [27]. Circularity of these kinds of irregular-shaped particles is in a range from 0.65 to 0.8. In this circularity range, virgin powder particles share a higher proportion than recycle powder particles, as shown in Figure 2c. 

In addition to the agglomerated particles, there are also elongated particles that originate from powder atomization and are rather difficult to remove during sieving. The elongated particles, e.g., Figure 3a3,b3, can be observed in both virgin and reused powders. They are typically characterized by circularity below 0.65 and limited in amount when compared to agglomerated and near-spherical ones.

The chemical changes for reused powder were revealed by XPS analysis. Although the L-PBF process operates under a protective atmosphere with argon, oxygen content is still relatively high (in the level of 1000 ppm) and leads to spatter oxidation, especially in the case of oxidation sensitive materials such as nickel-based superalloys [27,28,29].

Increased oxide formation in the case of the reused powder was observed by XPS, as seen in Figure 4. In general, a similar chemical composition of the powder surface was observed in both the as-received and reused powders representing powder alloy composition. The presence of a strong oxide peak in the case of both powders confirms that the powder is covered by an oxide film, predominantly formed by a thin Ni-oxide layer with the presence of the oxide islands because of higher thermodynamic stability as Cr, Al and Ti, which is typical for L-PBF of nickel-based alloys [27,28]. 

The most significant difference between the two powders is the surface enrichment of the reused powder in Cr, Al and Ti in an oxide state, indicating formation of oxide islands/patches with greater thicknesses, as well as larger coverage. An example of high-resolution spectra over the binding energy of chromium (Figure 5) clearly shows higher intensity of the Cr^3+^ in relation to the metallic Cr in the case of the reused powder compared to the virgin powder at all etch depths. It can be seen that at an etch depth of 2.6 nm, Cr is in a metallic state in the case of the virgin powder, whereas there is still some fraction of Cr in the oxide state in the case of the reused powder. Similar observations were made for Ti and Al.

### 3.2. Original Microstructure 

The E1 and E2 pristine microstructures from EBSD scans are summarized in Figure 6. As shown in the inverse pole figure (IPF) coloring maps (left column), the [001] direction of the grains is significantly parallel to the BD. In addition, columnar-shaped grain morphology is observed in both cases, where the longitudinal direction of the grains is closely parallel to the BD. The preferred grain orientation can be also confirmed from the pole figures (middle column); an obvious cube texture is obtained in both E1 and E2 samples. Here, the grain size is defined as the square root of the grain area. According to the grain size histogram (right column), both E1 and E2 samples follow log-normal distribution, where a slightly smaller average grain size of the reused sample (27.1 μm) is found compared to the virgin sample (28.7 μm).

Figure 7 presents SEM micrographs for the chemically-etched samples before testing. Two typical phases, γ matrix and γ′ precipitates, can be observed. The continuous phase, shown in light gray, is identified as the γ matrix and the discrete flower-shaped colony in dark gray is considered as γ′ precipitates. The flower-shaped γ′ precipitates are enriched in Al, Ti, and Ta, and depleted in Cr, Co and Mo compared to the γ matrix [19]. Elongated γ′ precipitates were observed in both specimens along the grain boundaries. In this study, the post-processing HIP temperature is above the γ′ solvus temperature, resulting in a fully dissolved γ′-phase. The elongated γ′ were formed during the cooling process from the super-solvus temperature. Based on the observation, the pristine microstructures γ matrix and γ′ precipitates are weakly dependent on the powder recycling.

### 3.3. Monotonic Tensile Properties

The engineering stress-strain curves of E1 and E2 specimens are shown in Figure 8a. No significant difference appears in the uniaxial monotonic tensile performance of the E1 and E2 specimens, and the engineering stress-strain curves of E1 and E2 are highly overlapped. Here, the 0.2% offset yield strengths are measured to be 965 MPa and 953 MPa, and the Young’s moduli are determined as 162 GPa and 167 GPa for E1 and E2, respectively. In addition, the E1 specimen shows a slightly higher tensile elongation at fracture (10.2%) than the E2 specimen (10.0%). The plots of work hardening rate (dσ/dε, here σ is the true stress and ε is the true strain) as a function of true strain of both E1 and E2 specimens are displayed in Figure 8b. The work hardening rate exhibits a distinction between E1 and E2, where the specimen fabricated by virgin powder shows a greater work hardening rate than the reused powder. Specifically, at around 4.5% true strain, the work hardening rate of E1 is 350 MPa higher than that of E2. However, from an overall point of view, the monotonic tensile performances of E1 and E2 are nearly consistent. 

Similar to the investigations of other studies [11,16,17], from a wide range of L-PBF processed materials including IN718, Ti-6Al-4V and AlSi10Mg alloy, and 17-4PH stainless steel, the powder reuse shows limited influence on the tensile performances. The monotonic tensile test was carried out as a guide for the stress-controlled fatigue test; therefore, only one specimen for each powder was tested. Furthermore, only powders with exactly the same usage and storage history were used for the printing of tested specimens to sustain the consistency of reused powder properties. Consequently, the amount of reused powder and the number of specimens were limited.

### 3.4. Fatigue Behavior

Fully reversed stress-controlled fatigue test results for all the specimens are listed in Table 3. Here, the stress-based approach is used to analyze the results based on Basquin’s equation, which is used for describing the trends in the stress/life behavior and for fitting the approximate straight line on the log-linear stress vs. fatigue life diagram [30]. The mathematical expression of these stress/life curves is written as:(2)$$\frac{∆\sigma }{2}=\sigma_{f}^{’}{\left(2N_{f}\right)}^{b}$$
where $\frac{∆\sigma }{2}$ is the stress amplitude $\sigma_{a}$, $\sigma_{f}^{’}$ is the fatigue strength coefficient and b is the fatigue strength exponent. Fatigue strength coefficient $\sigma_{f}^{’}$ is approximately equal to the true fracture stress for many metals. Fatigue strength exponent b is a negative value and a smaller value indicates a longer fatigue life. The fitted values for the E1 and E2 specimens are given in Table 4.

Figure 9 presents the fatigue life in a semi-log plot with the number of cycles to failure, $N_{f}$, on the X axis and stress amplitude, $\sigma_{a}$, on the Y axis. It is a Wöhler’s type stress versus life (S-N) figure and the dashed curves indicate the fitted fatigue data. As plotted, the fatigue life of both E1 and E2 specimens is generally dependent on the applied stress amplitude. For E1 specimens, it can be found that fatigue life gradually decreases with the increasing stress amplitude from 520 to 700 MPa. It should be noted that the E1 specimen subjected to the stress amplitude of 520 MPa runs outside of the pre-set fatigue life limit of 2,000,000 cycles. For E2 specimens, which were fabricated from the reused powder, the general behavior of reducing fatigue life as the function of increasing stress amplitude is still present. However, significant scattering of the fatigue life of E2 specimens at the higher stress amplitude regime ($\sigma_{a}$ > 600 MPa) can be found. 

It is noteworthy to point out that, at a stress amplitude of around 600 MPa, a crossover point exists for the E1 and E2 specimens. At lower stress, the E1 specimen shows a longer lifetime than the E2 specimen in the same test conditions. Whereas, on the higher stress regime, when the stress amplitude $\sigma_{a}$ = 645 MPa and 700 MPa, the E2 specimen shows obviously longer fatigue life than E1. However, a deviation in terms of scatter from the fitting curve is higher for E2 specimens than that for E1 specimens. The scatter depends on several factors, such as defects and microstructural variations in certain regions. The scatter is more obvious under higher stress amplitudes [31]. 

### 3.5. Fatigue Fracture

To further observe the fracture surface and understand the cracking mechanism, OM and SEM fractographic analyses were conducted to determine the crack initiation sites. The optical micrographs show wide views of the overall fracture surface, as shown in Figure 10. The fracture surfaces of E1 and E2 corresponding to the identical stress amplitude are gathered together. It should be mentioned that the E1 specimen under 520 MPa reached the runout limit without fatigue failure, and that the specific fracture surface in Figure 10 is created by a tensile overload afterwards.

In the high cycle fatigue (HCF) regime, surface-induced cracking or large inclusion cracking is commonly observed [32,33]. For example, crack initiation sites were found at the surface, at detectable shrinkage porosities and at internal large inclusions (in 100 μm scale) in modified 9Cr-1Mo ferritic steel [34]. In the current study, because of the L-PBF process with optimized printing parameters and post treatment including HIP, large inclusions are not expected. All crack initiation sites are typically observed close to the surface of the fatigue bar, marked as the white circles in Figure 10. After crack initiation, the fatigue crack propagates along the radial directions (green arrow line) of the initiation cracks, which is indicated by the blue and red shadows for E1 and E2, respectively. The crescent-shaped strip shadow indicates the area for crack initiation and propagation. Then, the final rupture region can be distinguished at the upper part of the micrographs. The moon-shaped dark zone beyond the strip is rather smooth compared with the initiation and propagation area. Such areas show a similar morphology to the run-out specimens.

In general, there are three typical types of fatigue crack initiation morphology under room-temperature HCF of nickel-based superalloys, namely pore-assisted crack initiation, lack of fusion-induced crack initiation and crystallographic facet crack initiation [35]. Pore assisted cracks can occur based on the size, location and proximity of the pore to a surface or another pore, which is usually accompanied by a fisheye-like pattern [36,37]. Lack of fusion-induced crack initiation is generally closely related to the welding processes, as well as the AM process. In this study, owing to optimal printing parameters and the following HIP process, specimens with crystallographic facet at or near the crack initiation area are much more common. Typical facets are shown in Figure 11 under SEM. The crystallographic facet is the frequently observed pattern in both E1 and E2 specimens under all the stress amplitudes. Although two other patterns have been observed by other researchers in fatigue tests for other AM materials [16,17,38], the difference in printing strategies, post-processing treatments and test conditions should be emphasized. Other fatigue traits can also be clearly observed from the fracture surface, i.e., secondary cracks, under high stress amplitude. 

## 4. Discussion

### 4.1. Powder Properties

The critical problem of powder reuse is powder degradation, which is a general term that includes particle size, shape, oxide surface, etc. One of the most commonly observed phenomena is the accumulation of the oxidized spatter particles in the powder after reuse [27,28,39]. Agglomerates of the powder particles can be also produced during powder reuse due to the redeposition of the molten spatter particles, either direct melt pool ejections or melted spatter particles by the laser beam, resulting in the formation of the particle agglomerate, as described in detail elsewhere [27,28]. Such agglomerates are often sieved out, as particle size after agglomeration is typically above the sieve mesh size (45 μm used in this study). Hence, the remaining issue is the spherical spatter particles that constitute the largest fraction of the spatter particles and are characterized by the high oxygen pick-up, determined by the oxygen potential in the process atmosphere [28], as well as the affinity of the material to the oxygen [40]. The higher oxygen level presents as Cr, Al and Ti oxide affiliated at the particle surface in this study, as indicated in Figure 5. Furthermore, such an increment in oxygen is generally found in almost all L-PBF materials after reuse [14,40,41,42]. 

In addition to the oxide, the size and shape of powder particles also slightly varies. Although any intuitive difference between the two powders from the morphological perspective is almost negligible, a statistical study on morphologies can determine the slight difference. Nevertheless, the drop in size and rise in circularity in this study were inconsiderable. Meanwhile, the trend of a lower distribution of irregular agglomerated particles (circularity ranges from 0.65 to 0.8 in Figure 2c) in reuse powder is clear. This might be explained by the sieving process because larger agglomerated particles over 45 μm would be excluded, but elongated particles would be included by chance. However, the difference in powder particle size and shape is still under debate. A slight coarsening of powder size between 50 µm to 100 µm after 14 times of reuse in IN718 was reported by Ardila et al. [43]. Size distribution of IN718 was reported to slightly increase after reuse and concentrated the mean value by Renderos et al. [14]. There was no significant change in size distribution, morphology and flowability for IN718 powders after a large number of cycles, as observed by Nandwana et al. [7]. As for circularity, both a decrease [41] and an increase [14] were observed for IN718. It is also worth mentioning that particle size and shape can also be affected by the printing component, and can sometimes play opposite roles during printing, as has been reported in several studies [41,44,45].

The variation in reused powder particles is difficult to predict, but may contribute to more unexpected defects and degradation of mechanical behaviors, especially the cyclic properties. Unfortunately, there is no standard or rule for powder reuse in the L-PBF process as this needs to be studied case by case.

### 4.2. Fatigue Cracking

The crystallographic facet is the microstructural features associated with crack initiation in the present study. EBSD was carried out for validation of the grain facet and the slip system from the cross sections close to the crack initiation sites. In Figure 12, the representative SEM micrographs of the cross-sectional area close to the crack initiation sites of E1 and E2 specimens tested at low stress condition ($\sigma_{a}$ = 545 MPa) and high stress condition ($\sigma_{a}$ = 700 MPa) are presented. The fatigue loading direction (LD) is parallel to the horizontal direction in the micrographs. The cracks initiated at the lower left corner and propagated upwards to generate the fatal fracture. In Figure 12a3–d3, the IPF coloring map superposed with the grain boundary (GB) plots from EBSD measurements of the same region in Figure 12a2–d2 are shown. The reference axis for the IPF coloring is parallel to the loading direction. The {111} plane traces are illustrated, which are the active slip systems for FCC materials as the IN738LC superalloy in this study. As shown in Figure 12a3–b3, the cross-section of the main fracture surface is well aligned and matched with the projection of {111} slip traces on the observation plane. Secondary cracks can also be observed under a higher stress amplitude of 700 MPa in Figure 12c3–d3. Generally, these secondary cracks still propagate within the {111} slip system, but not along the main crack propagation (111) plane.

According to the results from OM, the crystallographic facet is the most operative fatigue cracking initiation site. The dislocations move along closely packed planes near the crack initiation area; in this study, the {111} slip plane. Therefore, the fatigue life of the E1 and E2 specimens mainly depends on the dislocation slip and depends less on the presence of inclusions or pores at the levels studied in this work. The powder reuse effect on fatigue when considering defects is limited. 

## 5. Conclusions

The purpose of this study was to investigate the effect of powder reuse on mechanical properties of the nickel-based superalloy IN738LC, fabricated via L-PBF. The main conclusions were:

From the morphology and particle size, both virgin and reused powders are very similar, with a slight decrease in the mean particle diameter from 19.17 μm to 17.60 μm after reuse. The circularity of the powders is also similar in the range of 0.8. Particle agglomerates, as well as elongated particles, were observed in both cases, indicating that they originated from powder atomization. The only significant difference between the powders is increased coverage of the particle surfaces by oxide islands in the case of the reused powder. These oxides are predominantly formed by Cr-rich oxides with the presence of Al- and Ti-oxides. 

Powder reuse does not principally affect the printed microstructure, including grain size, morphology, cryptographic texture, γ matrix and γ′ precipitates. An almost negligible reduction of monotonic tensile strength and ductility is induced by powder reuse. Specimens fabricated from the reused powder showed lower yield strength, higher Young’s moduli and slightly lower tensile elongation compared with the specimens produced by the virgin powder.

Powder reuse has a weak negative influence on fatigue performance in terms of both fatigue life and scatter. A larger scattering of fatigue life occurs for specimens from the reused powder, especially at higher stress amplitudes. At the lower applied stress amplitudes, the specimens produced from reused powder display a shorter fatigue life. Fractured surfaces show a crescent-shaped strip at the crack initiation region at the surface of fatigue bars. 

No large-scale pore or lack of fusion were observed at the crack initiation sites. The operative crack initiation pattern was typically crystallographic facets. Both the primary and secondary cracks propagated along the dominant slip system {111}, which was consistent with the observation from the top view of the fracture surface.

## Figures and Tables

**Figure 1 materials-15-08925-f001:**
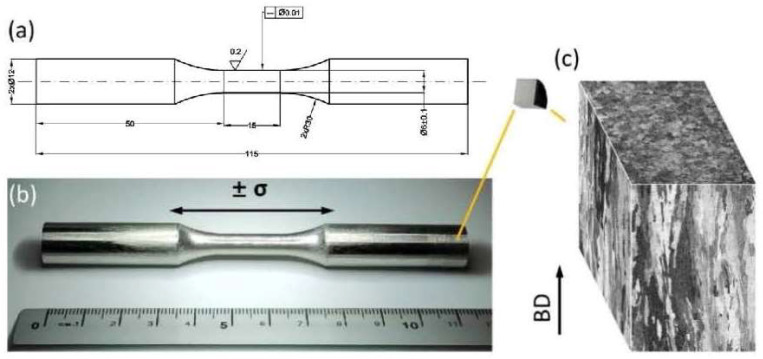
(**a**) The geometry (unit: mm) of the fatigue specimen used in this study; (**b**) the image of representative fatigue bar; (**c**) optical micrographs of the specimen before fatigue testing. The fatigue loading direction is perpendicular to the BD.

**Figure 2 materials-15-08925-f002:**
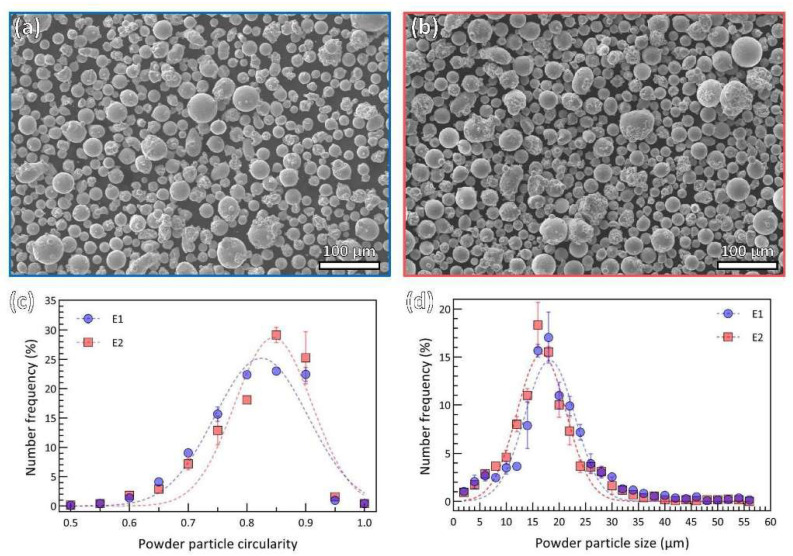
SEM micrographs showing the overview of powder particles (**a**) virgin powder (E1) and (**b**) reused powder (E2); powder particle size and shape distribution profile with number frequency as a function of powder particle size (**c**) and number frequency as a function of powder particle circularity (**d**) (dashed curves fitted by Gaussian distribution).

**Figure 3 materials-15-08925-f003:**
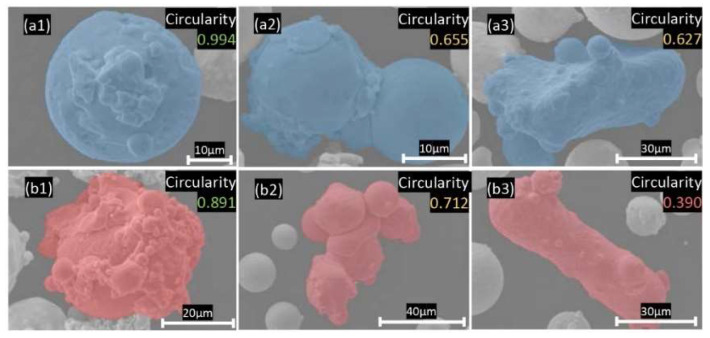
Typical morphology of non-spherical powder particles from SEM, (**a1**–**a3**) virgin powder with (**a1**) near spherical particle with satellites on surface; (**a2**) agglomerated particle; (**a3**) elongated particle, and (**b1**–**b3**) reused powder with (**b1**) near spherical particle with satellites on surface, (**b2**) particles agglomerate, (**b3**) elongated particle.

**Figure 4 materials-15-08925-f004:**
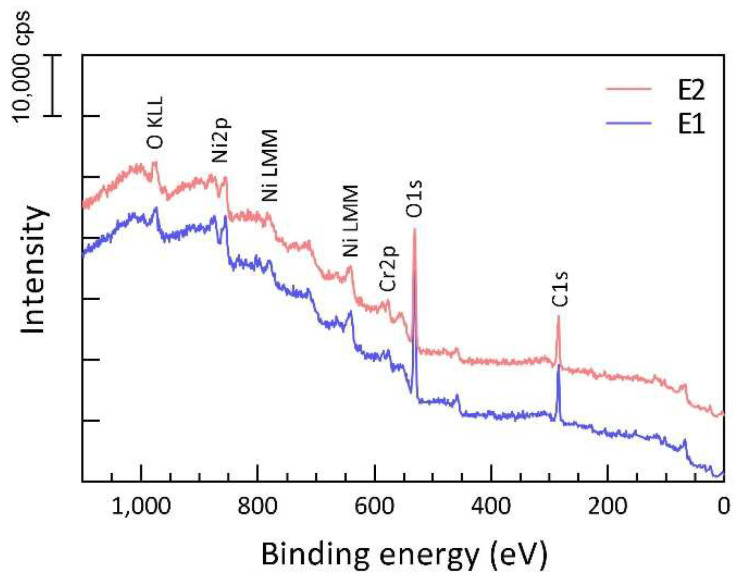
XPS survey spectra of powder E1 (blue) and E2 (red), indicating presence of surface oxide layer, covering powder in both cases.

**Figure 5 materials-15-08925-f005:**
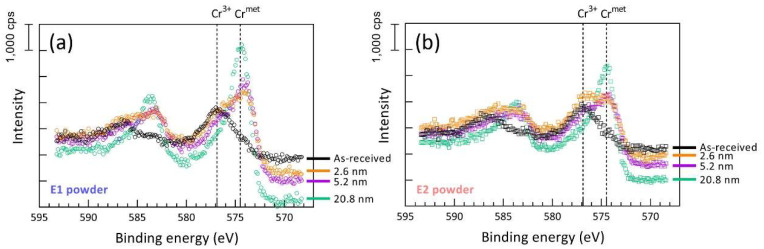
XPS narrow scans over the energy regions for Cr2p for (**a**) virgin and (**b**) reused powder, showing higher fraction of Cr in an oxide state in the case of the reused powder at all etch depth.

**Figure 6 materials-15-08925-f006:**
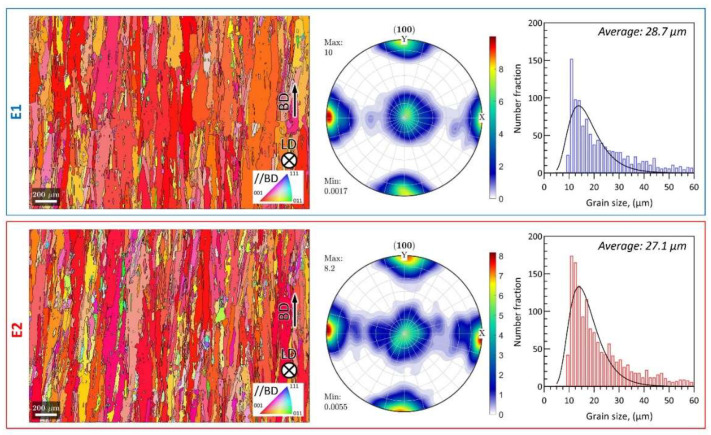
EBSD inverse pole figure coloring plots, grain size histograms and (100) pole figures of E1 and E2 samples before testing.

**Figure 7 materials-15-08925-f007:**
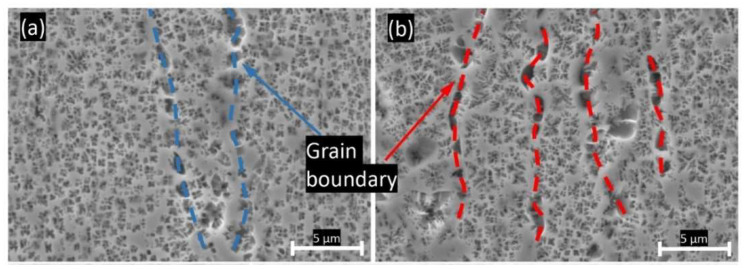
SEM micrograph of (**a**) E1 sample before test; (**b**) E2 sample before test.

**Figure 8 materials-15-08925-f008:**
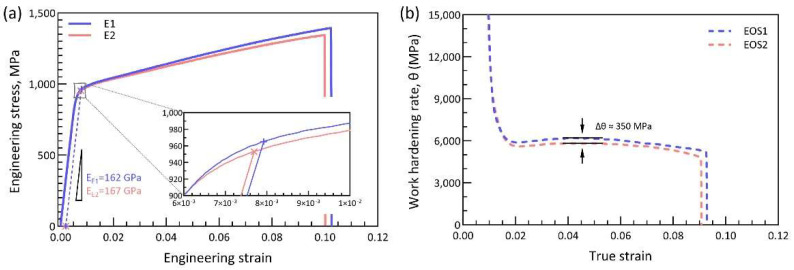
(**a**) Monotonic engineering stress-strain curve; (**b**) plots of work hardening rate as a function of true strain.

**Figure 9 materials-15-08925-f009:**
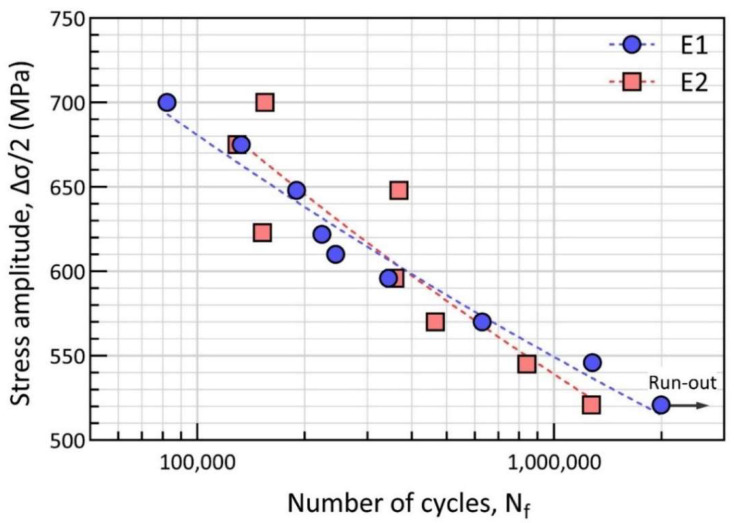
Fatigue stress/life (S-N) curve for E1 and E2 specimens.

**Figure 10 materials-15-08925-f010:**
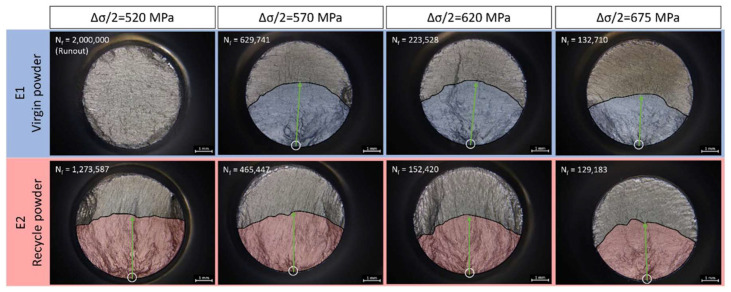
Fracture surface from optical micrographs (amplitude stress is 520 MPa, 570 MPa, 620 MPa, 675 MPa from left to right. Upper row is E1 and lower row is E2 specimens. The runout specimen was broken by the tensile test machine).

**Figure 11 materials-15-08925-f011:**
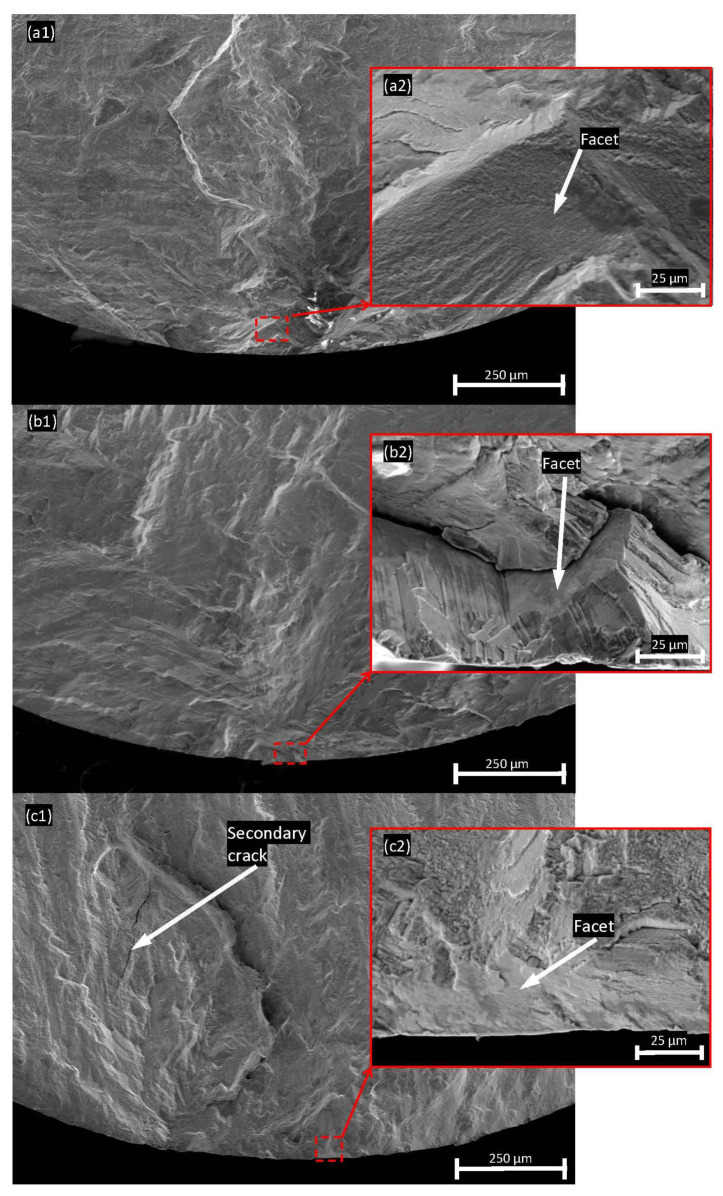
Typical SEM fractographies of fracture surface from E1 sample at (**a1**) 570 MPa; (**a2**) facet at near surface region; E2 sample at (**b1**) 545 MP; (**b2**) facet at near surface region and (**c1**) 700 MPa; (**c2**) facet at near surface region.

**Figure 12 materials-15-08925-f012:**
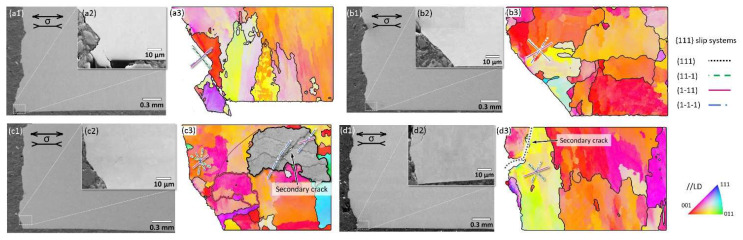
Secondary electron SEM micrographs and IPF coloring map superposed with grain boundary plots of the crack initiation regions of (**a1**–**a3**) E1 specimen and (**b1**–**b3**) E2 specimen, tested at $\sigma_{a}$ = 545 MPa; (**c1**–**c3**) E1 specimen and (**d1**–**d3**) E2 specimen, tested at $\sigma_{a}$ = 700 MPa. The fatigue loading directions are parallel to the horizontal direction. The family of {111} traces of crystallographic planes are indicated.

**Table 1 materials-15-08925-t001:** Optimized laser powder bed fusion process parameters used in this study.

Laser Power (W)	Scanning Speed (mm/s)	Hatch Distance (µm)	Layer Thickness (µm)
210	1750	50	20

**Table 2 materials-15-08925-t002:** Chemical composition of the IN738LC superalloy powder in the as-gas-atomized condition.

Element	Ni	Cr	Co	Ti	Al	W	Mo	Ta	Nb
wt. %	Bal.	17.2	9.4	3.6	3.5	2.6	1.9	1.74	0.9
**Element**	**C**	**Fe**	**Si**	**Zr**	**B**	**N**	**S**	**Mn**	**P**
wt. %	0.101	0.07	0.04	0.03	0.02	0.00564	0.00376	<0.005	<0.001

**Table 3 materials-15-08925-t003:** Fatigue testing results with various stress amplitudes (note for 610 MPa, only E1 was tested).

Stress Amplitude, *σ_a_* (MPa)	Sample	Cycles to Failure, *N_f_*
520	E1/E2	2,000,000 (Run-out)/1,273,587
545	E1/E2	1,282,237/839,941
570	E1/E2	629,742/465,447
595	E1/E2	344,026/357,996
610	E1	244,310
620	E1/E2	223,528/152,410
645	E1/E2	189,903/368,200
675	E1/E2	132,710/129,183
700	E1/E2	82,223/154,741

**Table 4 materials-15-08925-t004:** Fatigue strength coefficient and exponent of E1 and E2 specimen.

	Fatigue Strength Coefficient, $\sigma_{f}^{’}$ (MPa)	Fatigue Strength Exponent, b
E1	2065	−0.091
E2	2776	−0.113

## Data Availability

The data presented in this study are available on request from the corresponding author.

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
