# Peer review of "Cyclic Deformation Behavior of Additive-Manufactured IN738LC Superalloys from Virgin and Reused Powders"

_materials, 2022, doi:10.3390/ma15248925_

Round 1

Reviewer 1 Report

Dear authors,

1. The authors demonstrated the reuse of the powder of the alloy IN738LC widely used in the aeronautical industry. The alloy is used for long-life turbine work and guide vanes for marine and land-based industrial gas turbines, as well as corrosion-resistant turbine parts and components for aircraft engines. The reuse process must be taken into account as it deals with environmental issues. In addition to this fact, they carried out an excellent study showing the influence of intermetallic on the result of mechanical strength, comparing alloy powders without reuse and with reuse. In summary, there are the following points that highlight the study:

1)- the use of the Powder bed fusion (PBF) route

2)- reuse of powders (important environmental aspect),

3)- Demonstration of the influence of intermetallic on mechanical strength.

2. As I mentioned in the previous question, the authors made a great contribution to the environmental cause. In addition, we can highlight the issue of the PBF route itself:

Additive manufacturing has several processes to build objects in a completely different way than in the common market. The Powder Bed Fusion category is another option for 3D printing, which aims to form parts by adding materials. formed through the addition, layer by layer, of raw material in powder form, be it a polymer or metal, contributing to the reduction of costs, reduction of material waste and production of final components ready for use.

It is observed that there is a gap in the literature when correlating the PBF route and the understanding of the influence of intermetallic on mechanical strength - further considering the issue of reuse and non-reuse of powders.

3. The focal point that contributes to the uniqueness is the analysis of the morphology and the intermetallic in the mechanical resistance. I emphasize that our research group has been working in this line of research, that is, understanding the morphological influence and thermodynamic phases on mechanical strength and corrosion resistance, always looking for routes that allow the reuse of powders in order to reduce costs. and waste. In this sense, I made the authors' suggestion to investigate the corrosion resistance of the alloy they studied.

4.  Regarding the methodology used, I really liked the presentation and the experimental sequence. However, I must apologize because I forgot one detail: the authors should post their experimental errors.

5. The conclusion is consistent with the development of the work.

6. There is an acceptable number of references (44) that support the development of the work. I emphasize that, with the exception of Dr. Eduard Hryha, the others have only two self-citations. Regarding Dr. Hryha, there are 9 articles (which would be a high self-citation number). However, we need to highlight the following in this analysis: he published with other groups (there is a "mix" of researchers); in addition, we need to recognize that he is a researcher of excellence, being proven by a high h index (h index = 26)

7. Authors must include errors from experimental measurements (tables and figures).

Author Response

Hi,

Please see attached for response letter.

Best regards,

Jialiang

Reviewer 2 Report

This study has investigated the effects of fresh and reused powders on the microstructure and mechanical properties of additive-manufactured IN738LC superalloys, especially on the fatigue behaviours due to limited research, which is important to understand how the reused powder behaves from cost-effective and eco-friendly perspectives. Therefore, it is recommended to publish. However, it might be helpful if more information could be provided:

1.       In section 2.1 Materials, “After each build cycle, approximately 20% of the fresh powder was added to assure necessary amount of powder for the next building mission”, Is this only for making specimens E1 or for both E1 and E2?

2.       In section 2.2, “One specimen from each condition 117 was used for the tensile testing”, why was there only one specimen for each condition? How is the repeatability?

Author Response

(The authors gave the same response as above.)
